# Could Inflammatory Indices and Metabolic Syndrome Predict the Risk of Cancer Development? Analysis from the Bagnacavallo Population Study

**DOI:** 10.3390/jcm9041177

**Published:** 2020-04-20

**Authors:** Margherita Rimini, Andrea Casadei-Gardini, Alessandra Ravaioli, Giulia Rovesti, Fabio Conti, Alberto Borghi, Anna Chiara Dall’Aglio, Giorgio Bedogni, Marco Domenicali, Pierluigi Giacomoni, Claudio Tiribelli, Lauro Bucchi, Fabio Falcini, Francesco Giuseppe Foschi

**Affiliations:** 1Department of Oncology and Hematology, Division of Oncology, University Hospital Modena, 73828 Modena, Italy; 73828@studenti.unimore.it (M.R.); 79607@studenti.unimore.it (G.R.); 2Romagna Cancer Registry-Istituto Scientifico Romagnolo per lo Studio e la Cura dei Tumori (IRST) IRCCS, 47014 Meldola, Italy; alessandra.ravaioli@irst.emr.it (A.R.); lauro.bucchi@irst.emr.it (L.B.); fabio.falcini@irst.emr.it (F.F.); 3Department of Internal Medicine, Degli Infermi Hospital, 48018 Faenza, Italy; fabio.conti2@studio.unibo.it (F.C.); chiaradall@gmail.com (A.C.D.); fgfoschi@gmail.com (F.G.F.); 4Centro di Ricerca Biomedica Applicata (CRBA), Azienda Ospedaliero-Universitaria Policlinico S. Orsola-Malpighi e Università di Bologna, 40121 Bologna, Italy; alberto.borghi@auslromagna.it; 5Liver Research Center, Italian Liver Foundation, Basovizza, 34121 Trieste, Italy; giorgiobedogni@gmail.com (G.B.); ctliver@fegato.it (C.T.); 6Department of Medical and Surgical Sciences-DIMEC, S. Orsola-Malpighi Hospital, Alma Mater Studiorum-University of Bologna, 40121 Bologna, Italy; marco.domenicali@auslromagna.it; 7Department of Internal Medicine, Ospedale di Lugo, AUSL Romagna, 48022 Lugo, Italy; pierluigi.giacomoni@alice.it

**Keywords:** Inflammatory indices, NLR, SII, PLR, cancer incidence, breast cancer, lung cancer, colon cancer, Metabolic Syndrome

## Abstract

Background: Despite the robust data available on inflammatory indices (neutrophil lymphocyte ratio (NLR), platelet lymphocyte ratio (PLR), and systemic immune-inflammation index (SII)) and clinical outcome in oncological patients, their utility as a predictor of cancer incidence in the general population has not been reported in literature. Methods: The Bagnacavallo study was performed between October 2005 and March 2009. All citizens of Bagnacavallo (Ravenna, Emilia-Romagna, Italy) aged 30–60 years as of January 2005 were eligible and were invited by written letter to participate to the study. All participants underwent a detailed clinical history and physical examination following the model of the Dionysos Study. All blood values included in the analysis were obtained the day of physical examination. Cancer incidence data were obtained from the population-based Romagna Cancer Registry, which operates according to standard methods. The aim of this analysis was to examine the association between metabolic syndrome and baseline SII, NLR, and PLR levels, and the diagnosis of an invasive cancer in the Bagnacavallo study cohort. Results: At univariate analysis, metabolic syndrome was not associated with an increase of cancer incidence (HR 1.30; *p* = 0.155). High glucose (HR 1.49; *p* = 0.0.16), NLR HR 1.54, *p* = 0.002), PLR (HR 1.58, *p* = 0.001), and SII (HR 1.47, *p* = 0.006) were associated with an increase of cancer incidence. After adjusting for clinical covariates (smoking, physical activity, education, age, and gender) SII, PLR, and NLR remained independent prognostic factors for the prediction of cancer incidence. Conclusions: Inflammatory indices are promising, easy to perform, and inexpensive tools for identifying patients with higher risk of cancer in cancer-free population.

## 1. Background

In the era of immunotherapy, attention to the immune system and its interplay with cancer development and spread is growing day by day. Several data suggest that a systemic inflammatory response is associated with cancer development, angiogenesis promotion, apoptosis inhibition, and damage of DNA, thus resulting in cancer progression and metastasis [1]. In particular, during inflammation both cancer cells and cells of the cancer microenvironment, such as stromal cells, endothelial cells, or host infiltrating cells, release tumorigenic factors, including pro-inflammatory cytokines, pro-angiogenic and growth-promoting factors, anti-apoptotic and invasion-promoting factors, prostaglandins, and chemokines. Among them, IL-1, TNF, and IL-6 act as crucial mediators by activating NF-kB, STAT-3, and HIF-1, the key transcriptional factors that impact in any stage of tumorigenesis, from the initiation to the promotion, progression, and metastatization.

Therefore, on the one hand, the immune system detects and targets non-self agents (including mutated cells) by the activation of the innate immune system in the attempt to protect the organism. On the other hand, the immune cells involved in innate immune system are one of the major sources of the angiogenic, epithelial and stromal growth factors, which contribute to the neoplastic progression. Based on this relationship between systemic inflammation and cancer progression, several inflammation-based indices have been used as available and inexpensive surrogates of the immunological microenvironment and systemic inflammation. The main scores include the neutrophil lymphocyte ratio (NLR), platelet lymphocyte ratio (PLR), and systemic immune-inflammation index (SII), and they showed to have prognostic and predictive value in many types of invasive solid cancers, such as colorectal [2,3], gastric [4], hepatocellular [5,6,7], lung [8], and ovarian cancers [9].

Despite the robust data available on inflammatory indices (NLR, PLR, SII) as prognostic predictors for many types of oncological patients, their utility as predictors of tumor incidence in a cancer-free population has not been already reported in literature.

Concurrently, in the last decade the link between immune disequilibrium and MS has been highlighted [10]. MS is a severe health problem world-wide and clusters abdominal obesity, hypertension, dyslipidemia, and hyperglycemia, all these leading to insulin resistance and eventually diabetes mellitus, as well as non-alcoholic fatty liver disease.

However, MS is now recognized to be a pro-inflammatory condition [11,12], and adipose tissue is considered an active endocrine and paracrine organ [13]. Regarding MS, several studies prove that the pro-inflammatory state induced by adipokines production, insulin resistance, and oxidative stress could improve the risk of cancer [13,14,15] and cancer-related mortality [16]. The pathophysiology that underlines the connection between MS, inflammation, and cancer has not been elucidated yet. Some studies highlighted the role of adipose tissue hypoxemia as link between obesity, insulin resistance, and cancer development. The adipose tissue of obese patients acts as a cytokine deregulator by the stimulation of NF-kB, leading to a chronic inflammatory state, which, for its part, contributes to insulin resistance by interfering in the intracellular signaling cascade of insulin. The systemically elevated free fatty acids (FFA) and the decreased adiponectin levels found in the peripheral blood of people with MS further aggravate insulin resistance. The insulin resistance induces a relevant bioavailability of insulin-like growth factor-1 (IGF-1), and a consequent downstream activation of the PI3K/Akt pathway, which leads to a deregulated cell proliferation and inhibition of apoptosis [17,18,19].

Trying to deepen the significance of MS and its possible outcomes in the general population, Foschi et al. conducted the Bagnacavallo Study by collecting data about liver function in 3933 citizens from Bagnacavallo (a small town near Ravenna, in Emilia Romagna, Italy), in order to evaluate potential risk factors and health outcomes of fatty liver (FL) in general population. In particular, the authors recollected data about lipid assessment, liver function, and glucose metabolism, and a liver ultrasound was performed in order to investigate the presence of FL [18,20].

In this study Foschi et al. concluded that FL is highly prevalent in Northern Italy and more frequent in citizens with liver transaminases alteration compared to those with normal values (74% vs. 35%). They also found that all components of the metabolic syndrome were associated to FL independently of altered liver enzymes, gender, age, and alcohol consumption.

Starting from the same population of Bagnacavallo study, this study aims at evaluating the predictive value of metabolic syndrome and immune-inflammation indicators in order to identify the risk of cancer among a general population.

## 2. Materials and Methods

The Bagnacavallo study was performed between October 2005 and March 2009. All citizens of Bagnacavallo (Ravenna, Emilia-Romagna, Italy) aged 30–60 years as of January 2005 were eligible and were invited by written letter to participate in the study. Public encounters were also held to promote participation in the study [20].

All participants underwent a detailed clinical history and physical examination following the model of the Dionysos Study [21]. Weight and height were measured following international guidelines [22]. Body mass index (BMI) was calculated and classified following the NIH guide- lines [23]. Waist circumference (WC) was measured at the midpoint between the last rib and the iliac crest [24]. The performed blood tests included: (1) glucose; (2) triglycer-ides; (3) total cholesterol; (4) high-density lipoprotein (HDL) cholesterol; (5) low-density lipoprotein (LDL) cholesterol; (6) ALT; (7) AST; (8) gamma-glutamyl-transferase (GGT); (9) bilirubin; (10) hepatitis B surface antigen (HBsAg); and (11) antibodies against hepatitis C virus (anti-HCV).

All blood values included in the analysis were obtained the day of physical examination.

Cancer incidence data were obtained from population-based Romagna Cancer Registry, which operates according to standard methods [25]. We searched this ICD-10 code: C00–C43, C45–C96. The complete list of ICD10 codes and descriptions of tumor sites can be found in Appendix A.

The study was approved by the Ethical Committee of Area Vasta Romagna—IRST (reference number 112). All citizens gave written informed consent before they started with the clinical history, physical examination, and performed blood tests.

## 3. Statistical Analysis

The aim of this analysis was to examine the association between metabolic syndrome and baseline SII, NLR, and PLR levels and the diagnosis of an invasive cancer in the Bagnacavallo Study cohort.

Cancers were coded according to the International Code Disease classification version 10 (ICD-10). Eligible cancers had the following codes: C00–43, C45–96. The codes, by the tumor site, are shown in Appendix A

The cohort was cross-checked with the files of the cancer registry. Cancer-free survival (CFS) was calculated as the time from the date of the physical examination to the date of cancer diagnosis or death from any cause or the date of end of follow-up (31 December 2016), whichever occurred first. CFS curves were estimated using the Kaplan–Meier method and compared with the log-rank test. Hazard ratios (HR) and 95% confidence intervals (95% CI) were calculated using the Cox proportional hazards analysis starting from a crude model (with no covariates). The HRs were subsequently adjusted for patients age, gender, smoking, physical activity and education. Model 1 was adjusted for patient age. Model 2 was adjusted for patient age, gender, smoker, physical activity, and education. All variables were forced into the models.

The SII was calculated as platelet count × neutrophil count/lymphocyte count, NLR was obtained by dividing the absolute neutrophil count by the absolute lymphocyte count, and the PLR was calculated by as the ratio of the absolute platelet count to the absolute lymphocyte count.

MS was diagnosed using the harmonized international definition [26]. In detail, a large WC was defined as WC ≥ 102 cm in men and ≥ 88 cm in women; high triglycerides as triglycerides ≥ 150 mg/dL or use of triglyceride-lowering drugs; low HDL as HDL < 40 mg/dL in men and < 50 mg/dL in women or the use of HDL-increasing drugs; high blood pressure as systolic blood pressure ≥ 130 mmHg or diastolic blood pressure ≥ 85 mmHg or use or blood pressure-lowering drugs; high glucose as glucose ≥ 100 mg/dL or use of glucose lowering drugs; and MS as ≥ 3 of the above.

For continuous predictors, such as NLR, PLR, and SII, we identified the best cut-off using the ROC curve and Youden′s test [27]. STATA version 15 (StataCorp LP, College Station, TX, USA) was used for the statistical analyses. A *p*-value less than 0.05 was considered statistically significant.

## 4. Results

The dataset from the Bagnacavallo study included 4026 subjects that accepted to take part in the analysis. A total of 3810 of them were eligible for our analysis (Figure 1), and 884 (21.9%) presented metabolic syndrome.

At the time of database lock in 31 December 2016, after a median follow-up time of 10 years, 203 invasive cancer have been diagnosed (5.3%). Of these, 20.6% were breast cancers, 15.3% prostate cancers, 6.9% lung and colon rectal cancers, 6.4% hematological cancers, 4.9% cervical or endometrium or ovarian cancer, and 3.9% were gastric cancers.

The main characteristics of the study population were summarized in Table 1.

Figure 2 shows the ROC curve and area under curve level for continuous variables, such as NLR (**A**), PLR (**B**), and SII (**C**).

For NLR, the area under the curve in the ROC analysis was substantially low, 0.5506. Youden′s test was equal to 0.110 with a *p*-value of 0.0359 and returned 1.5 as a cutoff point value. For PLR, the area under the curve in the ROC analysis was 0.5420. Youden′s test was equal to 0.115 with a *p*-value of 0.036 and returned 110.6 as a cutoff point value. For SII, the area under the curve in the ROC analysis was 0.5438. Youden′s test was equal to 0.099 with a *p*-value of 0.0359 and returned 365,776 as a cutoff point value. So, SII ≥ 365,776, NLR ≥ 1.5 and PLR ≥ 110.6 were considered as elevated levels.

Table 2 shows the total number of subjects, the number, and the cumulative proportion of subjects with cancer at 10 years, and the crude hazard ratio for all potential cancer determinants.

With a univariate analysis, no statistically significant correlation was found between the diagnosis of metabolic syndrome and cancer incidence (HR 1.30; 95% CI: 0.91–1.86, log-rank test *p*-value = 0.155. An additional analysis was performed to evaluate the single components of metabolic syndrome and cancer incidence. With univariate analysis, Subjects with high glucose showed an increased risk of developing cancer (HR 1.49; 95% CI: 1.08–2.07 log-rank test *p*-value = 0.016) (Figure 3). No other correlation was found.

For inflammatory indices, univariate analysis, NLR, PLR, and SII showed to be a factor for increased risk of developing cancer (HR 1.54 95% CI 1.16–2.03 log-rank test *p*-value = 0.002; HR 1.58 95% CI 1.20–2.07 log-rank test *p*-value = 0.001; HR 1.47 95% CI 1.11–1.94 log-rank test *p*-value = 0.006; respectively) (Figure 3).

Table 3 shows the adjusted HR for cancer diagnosis associated with MS, NLR, PLR, and SII. Patients’ age was significantly associated with the HR for cancer diagnosis in all models. Gender, smoking, physical activity, and education, conversely, did not contribute significantly to their likelihood. An NLR ≥ 1.5, a PLR ≥ 110.6, and a SII ≥ 365,776 were positively associated with the adjusted HR for cancer diagnosis. A high level of glucose exhibited a positive association of borderline significance with the adjusted HR for cancer diagnosis.

## 5. Discussion

In this era of interest toward the immune system and its interplay with chronic ailments, including cancer, this is the first study performed in a healthy population that focuses on the possible predictive role of inflammatory indices and MS on cancer incidence.

Our analysis proves that inflammatory indices (NLR, PLR, and SII) are actual risk factors for cancer development in the general population. No data exists in the literature about the role of inflammatory indices in the general population, while several epidemiologic data have established an association between chronic inflammation and the development and progression of some kinds of cancers, including gastric, esophageal, colorectal, liver, pancreatic, bladder, and lung tumors [17,18,19,20,21,22,23,24,25,26,27,28,29].

The role of chronic inflammation as a promotor of carcinogenesis was first proposed by Rudolf Virchow in 1863, who detected leucocytes within cancer tissue [28,30]. Since Virchow’s observation, many studies were conducted with the aim to understand the complex crosstalk between inflammatory response and cancer.

Nowadays, inflammation is recognized as a hallmark feature of cancer development and progression [31]. The molecular mechanisms by which chronic inflammation drives carcinogenesis include production of cytokines, chemokines, and ROS (reactive oxygens species) that lead to the activation of transcription factors, such as NF-kB, STAT3, AP-1, and HIF-1a. All these transcription factors modulate the expression of genes involved in cell proliferation, invasion, metastasis, and angiogenesis [32,33,34].

Starting from these considerations, measurable blood parameters that reflect systemic inflammatory response have been incorporated in prognostic scores for several types of cancer [19]. NLR considers both the role of neutrophils and lymphocytes and reflects the altered myelopoiesis arising in cancer’s chronic inflammation. There is evidence that neutrophils and other immune cells, such as macrophages, have the capability to secrete cytokines and tumor growth promoting factors, including VEGF [35,36], HGF [37], IL-6 [38], IL-8 [39], matrix metalloproteinases [40], and elastases [40]. Moreover, it is proven that neutrophilia inhibits the cytolytic activity of immune cells, such as lymphocytes, activated T cells, and natural killers [40,41]. In contrast, lymphocytes can secrete some cytokines (IFN-gamma and TNF-alpha) involved in tumor growth control [42] and in systemic defenses. A recent meta-analysis including 100 studies has shown that a higher NLR is often associated with reduced overall survival, progression-free, and disease-free survival in solid cancers [43]. Small studies have even shown that chemotherapy can normalize elevated NLR and that patients with a normalization of NLR have better outcomes [44,45].

PLR is an inflammatory index that considers both the role of platelets and lymphocytes in the cancer-inflammatory response interplay. Evidence shows that tissue injury induced by cancer-related inflammation leads to platelet activation and aggregation, at the same time stimulating the coagulation cascade; thrombocytosis is very common in patients with solid tumors, indeed [46,47]. The key link between platelet production and inflammation seems to be IL-6: several studies have shown that IL-6 produced during inflammation stimulates the differentiation of megakaryocytes in platelets [48,49] and the production of thrombopoietin [50]. Moreover, platelets promote circulating tumor cells’ epithelial mesenchymal transition (EMT) [51]. A recent meta-analysis of 22 studies demonstrates an association between elevated PLR and poor prognosis in solid tumors on a total of 12,574 patients with solid tumors [52].

The SII, defined as platelets X neutrophils/lymphocytes, is a combination of the PLR and NLR and has been demonstrated to have a high prognostic value in some invasive cancers [53,54,55,56].

A previous meta-analysis performed on 13 studies indicated that elevated SII is associated with poor overall survival, progression free survival, cancer-specific survival, and time to recurrence [57].

Considering the link between chronic inflammation and cancer, and all the data supporting a prognostic role of inflammation biomarkers in many types of solid tumors, it is reasonable to assume that the inflammatory indices could be clinically useful to recognize an increased cancer risk in a cancer-free population. Our data support this hypothesis, but only conducting further investigations on a larger scale it will be possible to understand the best clinical use of inflammation parameters in a screening context.

In addition to cancer, another condition of chronic inflammation has been the object of study in recent decades: MS. Starting from the amount of evidence about the close relationship between obesity, insulin resistance, and innate immune activation, we investigated the possible connection between MS and cancer: unfortunately, our analysis showed no significant correlation between the diagnosis of metabolic syndrome and the development of malignancies.

In the last several years, many studies investigated the association between cancer risk and the different components of metabolic syndrome, but epidemiologic evidence is still scarce.

In a large population-based study conducted on a sample of 16,677 subjects on medications for hyperlipidemia, diabetes, and hypertension, an increased risk of pancreatic cancer in males and colorectal cancer in females was detected [58].

In a meta-analysis that evaluated 38,940 cancer cases, despite differences concerning sample characteristics and MS definition used in cohorts, the presence of MS has been shown to be in association with liver, colorectal, and bladder cancer in men and endometrial, pancreas, postmenopausal breast, rectal, and colorectal cancer in women [59]. However, if MS seems to constitute a risk factor for different types of cancer, no robust evidence exists to date to indicate whether MS leads to a higher risk of cancer than its individual components [60].

The biological mechanisms that link MS and cancer are far from being fully understood. It has been proposed that obesity and insulin resistance can constitute the main factors of the MS-cancer link. The insulin resistance stimulates the bioavailability of insulin-like growth factor-1 (IGF-1); the interaction between IGF-1 and its receptors induces the downstream activation of the Ras-Raf-MAPK and PI3K/Akt pathways, thus leading to increased cancer cell proliferation, and inhibition of apoptosis [17,18,19,61]. Furthermore, IGF-1 and hyperinsulinemia increase the bioavailability of sex hormones by inhibiting sex-binding globulin synthesis in liver and lead to growth and progression of hormone-dependent cancers [35]. Other links between MS and cancer involve an altered adipokines production and a local and systemic inflammation [36].

In our analysis, the reason behind the lack of correlation between MS, its components and cancer development could reside in the follow-up time. There are no data available concerning the time of exposure to MS that confers a higher risk of cancer onset, but it is likely that a follow-up time longer than the 3.5 years considered in our analysis might be necessary.

Furthermore, our dataset recollects data about 4026 citizens from the population of Bagnacavallo; thus, we considered a homogeneous population from the same geographic area, where citizens are supposed to have similar behavior and similar nutrition habits. On the other hand, only 203 citizens developed invasive cancers in our study follow-up, a small sample that does not allow further stratification analysis (for example, a stratification for type of cancer) that could make the correlation more evident and specific.

In conclusion, Inflammatory indices are promising, easy to perform and inexpensive tools for identifying patients with higher risk of cancer in general population; no significant correlation was found between MS, its components, and cancer incidence, but these results could be affected by the short time of follow up and small sample of invasive cancers developed. Inflammatory indices could be part of risk stratification criterions in the general population.

## Figures and Tables

**Figure 1 jcm-09-01177-f001:**
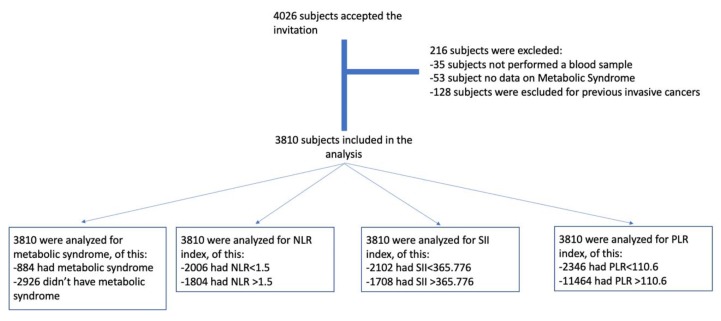
Flow chart of the study.

**Figure 2 jcm-09-01177-f002:**
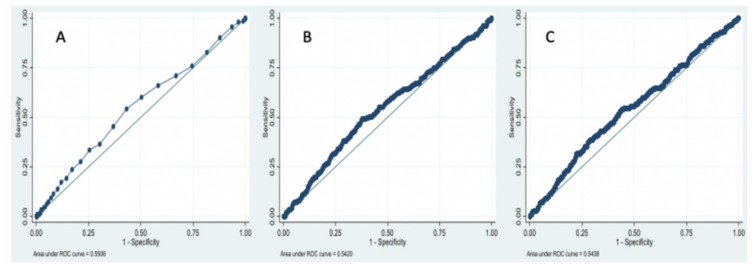
ROC curve of (**A**) NLR; (**B**) PLR and (**C**) SII.

**Figure 3 jcm-09-01177-f003:**
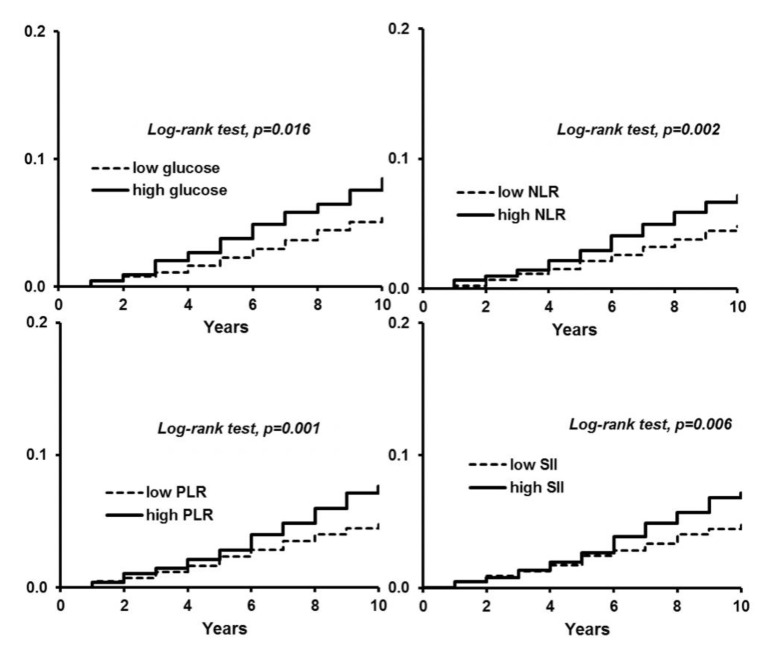
1–cancer-free survival and *p*-value for the logrank test for glucose, NLR, PLR, and SII.

**Table 1 jcm-09-01177-t001:** Main characteristics of the population study.

CHARACTERISTICS	PATIENTS, *n*. (%)
Gender	
Female	2008 (52.7)
Male	1802 (47.3)
Age, median (range)	47 (40–55)
Education	
≤Middle degree	1794 (47.1)
≥High degree	2016 (52.9)
Smoke	
Never	1889 (49.6)
Smoker/Former smoker	1910 (50.1)
Unknown	11 (0.3)
Physical activity	
No activity	2192 (57.5)
Activity	1617 (42.4)
Unknown	1 (0.1)
Waist circumference cm, median (range)	100 (93–107)
Triglycerides mg/dL, median (range)	96 (68–140)
HDL mg/dL, median (range)	61 (50–72)
Diastolic blood pressure (mmHg) median (range)	70 (70–90)
Systolic blood pressure (mmHg) median (range)	100 (90–130)
Glucose mg/dL, median (range)	89 (83–96)
NLR, median (range)	1.46 (1.14–1.87)
PLR, median (range)	101 (82–125)
SII, median (range)	345,662 (258,573–465,454)

**Table 2 jcm-09-01177-t002:** Total number of subjects, and subjects diagnosed with cancer.

	*n*.	*n*. Cancer	1-CFS, %	*p*-Value	Crude HR	95% CI	
Diagnosis of MS							
No	3261	167	5.6		1.00	reference	
Yes	549	36	7.8	0.155	1.30	0.91	1.86
High Waist circumference							
No	1209	56	5.0		1.00	reference	
Yes	2601	147	6.4	0.190	1.23	0.90	1.67
High Triglycerides							
No	2971	165	6.1		1.00	reference	
Yes	839	38	5.3	0.254	0.81	0.57	1.16
Low HDL							
No	3364	177	5.9		1.00	reference	
Yes	446	26	6.2	0.636	1.10	0.73	1.67
High Blood pressure							
No	3187	161	5.7		1.00	reference	
Yes	623	42	7.1	0.085	1.35	0.96	1.89
High Glucose							
No	3174	157	5.5		1.00	reference	
Yes	636	46	8.5	0.016	1.49	1.08	2.07
NLR ≥ 1.5							
No	2006	86	4.8		1.00	reference	
Yes	1804	117	7.2	0.002	1.54	1.16	2.03
PLR ≥ 110.6							
No	2346	103	4.9		1.00	reference	
Yes	1464	100	7.6	0.001	1.58	1.20	2.07
SII ≥ 365,776							
No	2102	93	4.9		1.00	reference	
Yes	1708	110	7.2	0.006	1.47	1.11	1.94

1–cancer-free survival, *p* value for the logrank test and crude hazard ratio with 95% confidence interval for all studied potential cancer determinants.

**Table 3 jcm-09-01177-t003:** Adjusted hazard ratio for cancer diagnosis associated with MS, NLR, PLR, SII, and with individual components of the metabolic syndrome.

	Model 1		Model 2	
	HR	(95% CI)	HR	95% CI
Diagnosis of MS				
No	1.00	reference	1.00	reference
Yes	1.01	(0.70–1.45)	0.98	(0.68–1.43)
High Waist circumference				
No	1.00	reference	1.00	reference
Yes	1.06	(0.77–1.44)	0.96	(0.68–1.35)
High Triglycerides				
No	1.00	reference	1.00	reference
Yes	0.73	(0.51–1.04)	0.74	(0.62–1.07)
Low HDL				
No	1.00	reference	1.00	reference
Yes	1.07	(0.71–1.61)	1.00	(0.66–1.53)
High Blood pressure				
No	1.00	reference	1.00	reference
Yes	0.93	(0.65–1.32)	0.91	(0.64–1.30)
High Glucose				
No	1.00	reference	1.00	reference
Yes	1.16	(0.83–1.62)	1.19	(0.84–1.68)
NLR ≥ 1.5				
No	1.00	reference	1.00	reference
Yes	1.52	(1.15–2.01)	1.53	(1.15–2.02)
PLR ≥ 110.6				
No	1.00	reference	1.00	reference
Yes	1.59	(1.21–2.09)	1.53	(1.18–2.07)
SII ≥ 365,776				
No	1.00	reference	1.00	reference
Yes	1.51	(1.15–1.99)	1.47	(1.11–1.94)

CI, confidence interval; HR, hazard ratio; MS, metabolic syndrome.

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
