# Peer review of "Could Inflammatory Indices and Metabolic Syndrome Predict the Risk of Cancer Development? Analysis from the Bagnacavallo Population Study"

_jcm, 2020, doi:10.3390/jcm9041177_

Round 1

Reviewer 1 Report

The paper is overall well structured despite a some comments. I have suggestion that could improve the paper:

  • Material and methods:
    • It would be better describe the study group and present the flowchart study in the materials and methods.
    • Describe NLR, PLR and SII and how was calculated.
    • Line 104, should be NLR.
  • Results:
    • The baseline characteristics of group need to be better described.
    • Describe multivariate analysis adjusted for age and gender. Show the analysis on table or figure.
    • In the statistical analysis you described, that you used the best cut-off using the Roc curve and Youden's test. Show the analysis on table or figure.

Author Response

The paper is overall well structured despite a some comments. I have suggestion that could improve the paper:

  • Material and methods:
    • It would be better describe the study group and present the flowchart study in the materials and methods.
    • RESPONSE: We had improve the material and methods and fig 1 (flow chart of the study)
    •  
    • Describe NLR, PLR and SII and how was calculated.

RESPONSE: We agree with the reviewer and we add in statistical analysis the description of NLR, PLR and SII

  • Line 104, should be NLR.
  • RESPONSE: We very thank the reviewer for this point. We have correct it
  • Results:
    • The baseline characteristics of group need to be better described.
    • RESPONSE: We completely agree with the reviewer and we add a table with baseline characteristics
    • Describe multivariate analysis adjusted for age and gender. Show the analysis on table or figure.

RESPONSE: The primary cox models have been adjusted for some factors: smoking, physical activity, education, age and gender. Smoking, physical activity and education.. We add this point in statistically chapter and we have changed all results chapter

  • In the statistical analysis you described, that you used the best cut-off using the Roc curve and Youden's test. Show the analysis on table or figure.
  • RESPONSE: We add supplementary figure 1 with the ROC curve

Reviewer 2 Report

At first sight, I thought this is an interesting study that investigated the relationship between inflammatory indicators and MS and incident cancer among the general middle-aged population. However, after I read it carefully, I found it has many problems in method description and results presentation.

Major comments

  1. Background:

First, the authors briefly described the application of inflammatory indicators in cancer diagnosis in the first third of the background, and point out that they may also have potential effects in predicting cancer incidence. However, the author did not go further with the impact of inflammatory indicators on the mechanism of cancer. Second, the authors then discussed the effect of MS on hepatocellular carcinoma. This confuses readers that authors would use liver cancer as the main outcome, but this was not the case. In the Methods and Results section, the authors discussed the development of overall cancer, breast cancer, gastrointestinal tumor, female cancer (this is a very vague definition), prostate cancer, and so on, but only did not describe anything about liver cancer. Third, the authors suggested that this study was conducted among a “healthy population”, for me, it is hard to say people with MS are healthy. The definition of study subjects is not clear. Last, it would be better to move line 72-77 to the Methods section, and line 78-81 seems unnecessary to the present study.

  1. Materials and Methods:

The authors made few informative descriptions of the selection flow for study participants, the measurement and definition for exposure, outcome and adjustment variables, and the follow-up duration. First, I strongly suggest authors move the flow-chart of participants to the Method section. Second, the details (date, measurement method, equipment, and contents, etc.) of all laboratory tests, examinations, questionnaires and so on should be introduced in detail (even in the appendix). Third, the ICD-10 code of cancer should be provided. Fourth, for follow-up, how did the authors deal with the dropped-out individuals? Fifth, it is unclear that the best cut-offs of NLP, PLR, and SII were used for what kind of analysis? Sixth, adjustment items for each model are not clear. Seventh, statistical significance level (two-sided, and differences at P<0.05 were accepted as statistically significant) should be specified in advance. Last, when did participants provide written informed consent?

  1. Results:

The Results section is not sufficiently structured. First, the baseline characteristics of participants should be provided. Second, HRs and hazard curves should be present separately. Third, for age-specified analysis, authors did not justify that why most age-specified results were use 40 years as a cut-off point. Fourth, the Figure 2 is quite difficult to understand. Authors mixed results of MS, results of NLR, SII and PLR, results according to glucose level, results only for women together. This makes me feel confused and painful when I was trying to understand the meaning of the graphs and compare them. Even not significant, all graphs for all exposure and subgroups should be present comparably. Authors should allow readers to judge the significance of the results through graphs on their own, instead of just providing what the authors think makes sense. Last, some confounding factors such as disease history should be adjusted.

  1. Discussion:

Considering the problems above, I did not go through the discussion part. Discussion should be considered after the above critical issue is resolved.

Minor comments

The term through whole paper should be checked and unified (e.g. “cancer” was used in most cases but in line 121, “tumor” was used, and in line 141, “invasive malignancies” was used; “inflammatory indexes”, “inflammatory indices” and “inflammatory indicators” were mixed).

Author Response

Major comments

  1. Background:

First, the authors briefly described the application of inflammatory indicators in cancer diagnosis in the first third of the background, and point out that they may also have potential effects in predicting cancer incidence. However, the author did not go further with the impact of inflammatory indicators on the mechanism of cancer. Second, the authors then discussed the effect of MS on hepatocellular carcinoma. This confuses readers that authors would use liver cancer as the main outcome, but this was not the case. In the Methods and Results section, the authors discussed the development of overall cancer, breast cancer, gastrointestinal tumor, female cancer (this is a very vague definition), prostate cancer, and so on, but only did not describe anything about liver cancer. Third, the authors suggested that this study was conducted among a “healthy population”, for me, it is hard to say people with MS are healthy. The definition of study subjects is not clear. Last, it would be better to move line 72-77 to the Methods section, and line 78-81 seems unnecessary to the present study.

RESPONSE: Thank you for the suggestions. We agree with your considerations, and we made some changes in the text. First of all, in the first third of the background we added a paragraph with the aim to better explicate the impact of inflammatory cells (which put in relation with each other constitute the inflammatory indicators such as NLR, PLR, SII) on the mechanism of cancer. Concerning the second point, we agree with your consideration about the confounding effect of the discussion of the effect of MS on hepatocarcinoma: it has been written as a completion of the issue, but it is an unnecessary date and HCC is not the main outcome of the work. So we removed the relative period. Concerning the third point, we obviously used the term “healthy population” in reference to cancer-free people. Nevertheless, we  agree that it can be confusing, or even incorrect. 

  1. Materials and Methods:

The authors made few informative descriptions of the selection flow for study participants, the measurement and definition for exposure, outcome and adjustment variables, and the follow-up duration. 

First, I strongly suggest authors move the flow-chart of participants to the Method section. 

RESPONSE: We move the flow chart in the method section

Second, the details (date, measurement method, equipment, and contents, etc.) of all laboratory tests, examinations, questionnaires and so on should be introduced in detail (even in the appendix). 

RESPONSE: We completely agree with the reviewer and we have added in material and methods all missing data.

Third, the ICD-10 code of cancer should be provided. 

RESPONSE: We completely agree with the reviewer and we have added in material and methods all missing data.

Fourth, for follow-up, how did the authors deal with the dropped-out individuals? 

RESPONSE: We censored at the time that the individuals changed the residence outside of Bagnacavallo city. We added in statistically analysis this point

Fifth, it is unclear that the best cut-offs of NLP, PLR, and SII were used for what kind of analysis? 

RESPONSE: We completely agree with the reviewer and we apologized with her/his for this mistake. We added in statistically analysis this point

Sixth, adjustment items for each model are not clear. 

RESPONSE: We completely agree with the reviewer and we apologized with her/his for this mistake. We added in statistically analysis this point

Seventh, statistical significance level (two-sided, and differences at P<0.05 were accepted as statistically significant) should be specified in advance. 

RESPONSE: We added this point in statistically analysis.

Last, when did participants provide written informed consent?

RESPONSE:We add in the material and methods that the participants written informed consent before they started with clinical history, physical examination and performed blood tests.

  1. Results:

The Results section is not sufficiently structured. 

First, the baseline characteristics of participants should be provided. 

RESPONSE: We add a table with baseline characteristics of participants.

Second, HRs and hazard curves should be present separately. 

RESPONSE: We separate univariate analysis from multivariate analysis and we changed all results chapter.

Third, for age-specified analysis, authors did not justify that why most age-specified results were use 40 years as a cut-off point. 

RESPONSE: We used the age cut-off of 40 years in view of the low risk of cancer among people under that age and for the resulting instability of estimates (further exacerbated by the small size of the younger population). We add this point statistically chapter.

Fourth, the Figure 2 is quite difficult to understand. Authors mixed results of MS, results of NLR, SII and PLR, results according to glucose level, results only for women together. This makes me feel confused and painful when I was trying to understand the meaning of the graphs and compare them. Even not significant, all graphs for all exposure and subgroups should be present comparably. Authors should allow readers to judge the significance of the results through graphs on their own, instead of just providing what the authors think makes sense. 

RESPONSE: We completely agree with the reviewer. We delete the fig 2 and we add new 4 figure. Please see the results section.

Last, some confounding factors such as disease history should be adjusted.

RESPONSE: In the first paper the cox models was adjusted for some factors: smoking, physical activity, education, age and gender. Smoking, physical activity and education We add this point in statistically chapter and we rewritten all results chapter.

  1. Discussion:

Considering the problems above, I did not go through the discussion part. Discussion should be considered after the above critical issue is resolved.

RESPONSE: We rewritten all results chapter and actually we think that is more clear. Between this version and first version the results have not changed because we had already done all the required analyzes but as the reviewer perfectly wrote they were not clear

Minor comments

The term through whole paper should be checked and unified (e.g. “cancer” was used in most cases but in line 121, “tumor” was used, and in line 141, “invasive malignancies” was used; “inflammatory indexes”, “inflammatory indices” and “inflammatory indicators” were mixed).

RESPONSE: Thank you Reviewer for your annotations, I agree with what you suggested. I’ve checked the paper and homogenize the terminology.

Round 2

Reviewer 1 Report

Dear Author,

Thank you for your change. I have minor suggestion. I think you can add information about area under curve (AUC) and results of Youden test and statistical significance in supplementary figure 1 with the ROC curve.

Author Response

Thank you for your change. I have minor suggestion. I think you can add information about area under curve (AUC) and results of Youden test and statistical significance in supplementary figure 1 with the ROC curve.

REPLY: We very thank the reviewer for this point; We add a new ROC curves in the text and we have added the results of Youden test.

Reviewer 2 Report

I can feel that authors have tried to revise their manuscript, however, there still some critical problems left.

Abstract

The abstract needs to be revise a lot. The Methods are too simple to provide enough information (details of subjects, follow-up duration, exposure, outcome, and statistical methods are necessary), and the Results need to be presented with HRs (95% CIs).

Background

The authors still left considerable contents that involved association between MS and the NAFLD. However, the NAFLD never appealed again in the rest of the manuscript. I prefer to learn more about the inflammatory indexes, MS and cancer incidence (and the mechanism and hypothesis related to the present study)

Methods

First, the date (duration) of baseline survey is not clear.

Second, the ICD-10 code should be given according to different cancer, respectively.

Third, the definition of MS and the criteria of each components of MS should be given in the manuscript (not just a citation).

Fourth, why the number of participants used for different exposures are different, although the authors have excluded individuals with missing data?

Fifth, the baseline characteristics should be given according to different exposure groups (e.g. participants with MS vs. participants without MS; participants whose NLR<1.6 vs participants whose NLR≥1.6, etc.), and it is better to put baseline characteristics in the Results section.

Sixth, the categories of covariates should be given in the statistical analysis part. For Roc curve, the outcome is unknown.

Results

First, the authors misunderstood the meaning of univariate analysis here. They used the Cox proportional hazards model, the phrase “univariate analysis” is misleading. I think they may want to say the “crude model”, comparing to the “multivariate adjusted Cox model” later in the text.

Second, the HRs (95% CI) should be given, not just the p values. To my point, the results of multivariate adjusted Cox model are more important than those of the crude model. Therefore, the presentation of the results is not very appropriate.

Third, all tables also need to be given with 95% CIs.

Fourth, the results regarding inflammatory indexes and different types of cancer have been given incorrectly.

Overall, there are still words which are not unified throughout the manuscript and redundant text in the Methods and Results section. I strongly recommend the authors refer to previous studies which have similar design to reconstruct their manuscript.

I will go through the Discussion section after the above issues are addressed.

Author Response

Abstract

The abstract needs to be revise a lot. The Methods are too simple to provide enough information (details of subjects, follow-up duration, exposure, outcome, and statistical methods are necessary), and the Results need to be presented with HRs (95% CIs).

REPLY: We have written all the abstract.

Background

The authors still left considerable contents that involved association between MS and the NAFLD. However, the NAFLD never appealed again in the rest of the manuscript. I prefer to learn more about the inflammatory indexes, MS and cancer incidence (and the mechanism and hypothesis related to the present study)

REPLY: Thank you for the considerations, I agree with that, so I’ve reported some more biologic details about the link between inflammation, MS and cancer, I don’t have reported more data because the literature about the theme will be deeply discussed in the discussion part.

Methods

First, the date (duration) of baseline survey is not clear.

REPLY: As reported in the first sentence of materials and methods the study was performed between October 2005 and March 2009

Second, the ICD-10 code should be given according to different cancer, respectively.

REPLY: We add supplementary table 1 with all ICD-10 code that we used.

Third, the definition of MS and the criteria of each components of MS should be given in the manuscript (not just a citation).

REPLY: We completed agree with the reviewer and we added in Statistical Analysis this point.

Fourth, why the number of participants used for different exposures are different, although the authors have excluded individuals with missing data?

REPLY: we have redone the figure 1. We excluded 216 subjects for these rason:

-35 subject not performed a blood sample

-53 subject no data on Metabolic Syndrome

-128 subjects were excluded for previous invasive cancers

Fifth, the baseline characteristics should be given according to different exposure groups (e.g. participants with MS vs. participants without MS; participants whose NLR<1.6 vs participants whose NLR≥1.6, etc.), and it is better to put baseline characteristics in the Results section.

Sixth, the categories of covariates should be given in the statistical analysis part. For Roc curve, the outcome is unknown.

Results

First, the authors misunderstood the meaning of univariate analysis here. They used the Cox proportional hazards model, the phrase “univariate analysis” is misleading. I think they may want to say the “crude model”, comparing to the “multivariate adjusted Cox model” later in the text.

Second, the HRs (95% CI) should be given, not just the p values. To my point, the results of multivariate adjusted Cox model are more important than those of the crude model. Therefore, the presentation of the results is not very appropriate.

Third, all tables also need to be given with 95% CIs.

Fourth, the results regarding inflammatory indexes and different types of cancer have been given incorrectly.

Overall, there are still words which are not unified throughout the manuscript and redundant text in the Methods and Results section. I strongly recommend the authors refer to previous studies which have similar design to reconstruct their manuscript.

 REPLY: For all points before we have redone all section of the paper (MATERIALS AND METHODS, Statistical Analysis and results). We hope that this new version has improved and we thank the reviewer for her/his suggestions.

I will go through the Discussion section after the above issues are addressed.